# Cellular Internalization of Beta-Carotene Loaded Polyelectrolyte Multilayer Capsules by Raman Mapping

**DOI:** 10.3390/molecules25071477

**Published:** 2020-03-25

**Authors:** Loredana F. Leopold, Oana Marișca, Ioana Oprea, Dumitrița Rugină, Maria Suciu, Mădălina Nistor, Maria Tofană, Nicolae Leopold, Cristina Coman

**Affiliations:** 1Faculty of Food Science and Technology, University of Agricultural Sciences and Veterinary Medicine, Calea Mănăștur 3-5, 400372 Cluj-Napoca, Romania; loredana.leopold@usamvcluj.ro (L.F.L.); ioana.oprea@usamvcluj.ro (I.O.); madalina.nistor@usamvcluj.ro (M.N.); maria.tofana@usamvcluj.ro (M.T.); 2Faculty of Physics, Babeș-Bolyai University, Kogalniceanu 1, 400084 Cluj-Napoca, Romania; nicolae.leopold@phys.ubbcluj.ro; 3Faculty of Veterinary Medicine, University of Agricultural Sciences and Veterinary Medicine, Calea Mănăștur 3-5, 400372 Cluj-Napoca, Romania; dumitrita.rugina@usamvcluj.ro; 4National Institute for Research and Development of Isotopic and Molecular Technologies, Donath 67-103, 400293 Cluj-Napoca, Romania; maria.suciu@itim-cj.ro

**Keywords:** beta-carotene, Raman mapping, polyelectrolyte multilayer, microcapsules, encapsulation

## Abstract

Raman mapping is becoming a very useful tool in investigating cells and cellular components, as well as bioactive molecules intracellularly. In this study, we have encapsulated beta-carotene using a layer-by-layer technique, as a way to enhance its stability and bioavailability. Further, we have used Raman mapping to characterize the as-obtained capsules and monitor their uptake by the human retinal epithelial D407 cells. We were able to successfully map the beta-carotene distribution inside the capsules, to localize the capsules intracellularly, and distinguish between capsules and other cellular components.

## 1. Introduction

An efficient tool to preserve bioactive compounds is microencapsulation [1]. Its multiple facets provide a reliable method to use in food and health industry for safe delivery of unstable extracts, vitamins, nutraceuticals, or drugs [2]. The beauty of microencapsulation resides in its ability to custom design the process for its desired application. For instance, one can protect the active compounds from oxygen, heat or light degradation, or it can encapsulate poor water-soluble compounds, or it can preserve its antioxidant activity, or just mask some organoleptic problems such as bad taste or odor [3].

Layer-by-layer assembly is one of the mostly used encapsulation techniques [4]. It is an intelligent system based on coating of a “sacrificial” template with electrostatic interacting polymers while adding layers in a controlled manner or other adjuvants, such as dyes, plasmonic nanoparticles, magnetic nanoparticles, thus tailoring the capsule towards its desired application. During the core formation, the bioactive compound is added, and it is passively trapped in the core pores and subsequently protected by addition of coating layers. Such a delivery system gained a high scientific interest due to its targeted delivery, ease of preparation, tunable characteristics, biodegradability, and biocompatibility [5].

Beta-carotene is one of the mostly known carotenoids with a high provitamin A activity whose low bioavailability, poor water solubility, and environment induced instability makes it a good candidate compound for micro-encapsulation [6]. Moreover, beta-carotene was extensively studied for its health benefits which comprise of vitamin A precursor, high antioxidant activity, and direct correlation with low incidence of heart disease and macular degeneration.

It was found that encapsulation of beta-carotene enhances its bioavailability while preserving its active form [7]. One of the most popular encapsulation techniques employed for beta-carotene is spray-drying [8]. In this technique, the wall material is very important, thus there are a multitude of studies on different wall types such as polysaccharides or natural polymers. Other encapsulation techniques use lipid-based particles and emulsions [9,10].

Raman mapping/imaging is one of the characterization techniques used more and more in cellular imaging [11]. This method provides the advantage of being noninvasive, label-free, highly specific, and spatially accurate [12].

In this study, we decided to employ a layer-by-layer (LBL) technique using polyelectrolyte polymers to encapsulate beta-carotene. Beta-carotene was successfully entrapped in PSS/PAH (poly (sodium 4-styrene sulfonate) and poly (allylamine hydrochloride)) microcapsules. The intracellular fate of LBL polyelectrolyte capsules was previously investigated and it was found that such microcapsules reach the cell cytosol by endocytotic pathways [13]. The capsules formed in our study were studied through Raman spectroscopy in combination with multivariate statistical analysis. Furthermore, their uptake and stability were studied in human retinal epithelial D407 cells through Raman mapping.

## 2. Results

Prior to cell-microcapsules mapping, empty and loaded polyelectrolyte multilayer (PEM) capsules were characterized, and Raman screened to establish the distinct spectral features of the components.

### 2.1. Capsules Characterization

Post synthesis, the microcapsules were characterized by dynamic light scattering (DLS) and zeta-potential measurements to assess the capsules size, distribution, uniformity, and surface charge. A summary regarding these properties is presented in Figure 1c. These physical-chemical properties are important in order to predict the cell–microcapsule interaction. 

Due to the nature of the microcapsule’s synthesis process, one can choose how many layers the wall has, thus how big the capsule is, and also what type of charge the exterior polymeric layer has, positive or negative. In this particular case, we started with the negative polymer PSS and ended with the positive one PAH. The reasoning behind choosing six layers is because they would offer enough protection against beta-carotene leakage while maintaining the size of the capsules in the lower micro-range. Furthermore, the last layer is positively charged in order to promote an electrostatic interaction with the cellular membrane which is negatively charged thus enhancing the microcapsule’s cellular uptake. According to the HPLC analysis, the encapsulation efficiency of beta-carotene is about 93%.

According to the DLS and zeta-potential measurements, the average of three separate read-outs indicate that the beta-carotene microcapsules have a hydrodynamic diameter of approximately 1.3 µm and a surface charge of +16.8 mV (Figure 1c). Analysis of TEM images (not shown here) indicate a size distribution of the microcapsules is in the 1–5 µm range. SEM analysis showed uniform round capsules with a finely rough surface (Figure 1b).

Figure 1a shows the Raman fingerprint of the nonloaded microcapsules, beta-carotene loaded microcapsules, and beta-carotene powder. By analyzing the beta-carotene Raman fingerprint peaks measured from the beta-carotene powder, two major peaks arise. The first one at 1155 cm^−1^, corresponds to C-C stretching and the second one at 1512 cm^−1^ corresponds to collective C=C stretching of the conjugated unsaturated backbone of beta-carotene. A third smaller peak at 1006 cm^−1^ is attributed to C-CH_3_ stretching [14,15,16]. The beta-carotene loaded capsules exhibit a Raman profile very similar to the Raman spectrum of beta-carotene powder. Basically, the same three vibrations of beta-carotene are visible in the characteristic Raman spectrum of beta-carotene-loaded capsules [17]. It is noticeable that the polymers that make up the capsule wall do not contribute to the beta-carotene loaded capsules Raman signal, beta-carotene being the major contributor to the signal. 

The Raman spectrum of nonloaded capsules was also recorded, which indicates that the characteristic peaks for the polymers PSS and PAH (the polymers found in the capsule’s wall) are at 1126 cm^−1^ which corresponds to the C-C stretching vibration from the polymer chain, and 1597 cm^−1^ which is associated to the C=C aromatic ring quadrant stretching from the PSS polymer [18,19,20].

After this basic Raman analysis of the microcapsules, the focus shifted further analyzing a multitude of Raman spectra collected at different spatial points in the sample and to transform these into score plots using relevant principal component analysis (PCA). PCA results in several significant principal components (PCs) which carry appropriate information on the analyzed samples [21]. PCA of the recorded Raman spectra provided false color score plots (maps) and their corresponding loading plots. Blue color is arbitrarily attributed to most negative values in the loading plots, while red corresponds to most positive values in the loadings. The maps are obtained by integration over the whole wavenumber range recorded.

### 2.2. Raman Mapping

Figure 2 shows the Raman map of nonloaded PEM capsules and the corresponding loading vector associated with the first PC. The loading vector of PC1 contains the characteristic Raman bands of PSS/PAH, at 1126 and 1597 cm^−1^. The PCA score and loading plots corresponding to the first ten PCs are represented in Appendix A.

The color map in the score plot depicts low intensities of the loading vector in each pixel with blue and high intensities of the loading vector with red. Accordingly, the pixels that contain the capsules show a high intensity of 1126 and 1597 cm^−1^ and consequently are assigned the red color, while the background shows practically no Raman signal and is assigned blue color.

When comparing the microscopy image of the capsules, it can be observed that the Raman map replicates the image and allows easily discrimination of the three capsules analyzed. This is proof that the Raman mapping can be used as a tool to selectively image a 3D structure based on the Raman fingerprint of the structure. 

A similar analysis was performed on beta-carotene loaded PEMs, shown in Figure 3. For the beta-carotene loaded PEM capsules, the loading vector corresponding to PC1 shows the characteristic vibrations from beta-carotene. Therefore, the intensity of the PC1 loading vector in each pixel of the mapped sample forms the corresponding score plot. The PCA score and loading plots corresponding to the first ten PCs are shown in Appendix A.

As before, red color signifies areas in the sample with a high Raman intensity of beta-carotene, whereas blue color is assigned to areas in the mapped sample with no Raman bands from beta- carotene. Noticeably, the zones abundant in beta-carotene are in the center of the capsules, which indicates that beta-carotene is successfully entrapped in the capsules and no leaks are detected with Raman spectroscopy. 

The first PC of the Raman spectra PCA provides a Raman map with very good spatial resolution, enabling the identification of the beta-carotene loaded PEM capsules. The corresponding PC1 loading plot shows the two beta-carotene marker bands at 1157 and 1520 cm^−1^, as well as the beta-carotene typical fluorescent background.

Before any in vitro Raman mapping was performed, there was a need to assure that the PEM capsules are internalized by the D407 cells. The validation was obtained by TEM imaging of D407 cells. As it can be seen in Figure 4 in the TEM micrograph, three microcapsules are internalized in the D407 cell cytoplasm.

The next step further in validating the Raman mapping specificity is to evaluate the ability to discriminate the compounds of interest in an in vitro setting. The D407 cells were exposed to empty PEMs for 24 h and then they were imaged and analyzed via Raman spectroscopy. Figure 5 presents a collection of maps corresponding to the most significant PCs obtained by PCA of the Raman spectra registered. 

The PC2 loading plot in Figure 5 shows positive bands due to MgF_2_ (used as substrate for the cells) at 406 cm^−1^, and a superposition of bands due to the polymer (1126, 1598 cm^−1^) and typical cell Raman bands, the most evident being at 1001, 1446, 1654 cm^−1^. The band at 1001 cm^−1^ is assigned to aromatic ring stretching of phenylalanine amino acid from proteins, the 1446 cm^−1^ vibration represents the C-H bending of lipids, and 1654 cm^−1^ contains mixed contribution from amide I band of proteins and water [22,23,24]. The PC2 score plot depicts the internalized capsule and the shape of the cell.

However, the cell shape can be observed more clearly in the score plot of PC3, the loading of PC3 showing typical cell Raman bands. The capsule shape can be recognized also in the score plot of PC3. The negatively correlated MgF_2_ Raman band with the cell Raman bands generates a dark blue region in the score plot, in the close vicinity of capsule.

The shape in PC4 score plot is attributed to the capsule, as also indicated by the characteristic bands in the loading plot. The negative peaks in PC4 correlate with the PSS/PAH signature peaks corresponding to the C-C stretching vibrations at 1126 and 1597 cm^−1^ vibration associated with the aromatic ring quadrant [20]. Thus, PC4 map provides evidence that the PEM capsules are internalized in the cell and they can be easily identified among other cell structures. 

Cellular components can be recognized in the score plot of PC5. Typical cellular bands of phenylalanine, lipids, amide I band of proteins are present at 1003, 1446, and 1654 cm^−1^, respectively, but also the 788 cm^−1^ positive band, which is assigned to the PO_4_ vibration of DNA (O-P-O stretching), the high DNA content being typical for the cell nucleus [24] shown in red color in the score plot. The negative peak at 1446 cm^−1^ in PC5 score plot can be assigned to C-H bending vibration of lipids; depicting these cytoplasmic components in blue color, since lipid synthesis is mainly located in the endoplasmic reticulum [23]. The spectral fingerprint representing the DNA is negatively correlated with the one representing the C-H vibration, so nucleus is shown in red and the cytoplasm in blue. The PCA score and loading plots corresponding to the first ten PCs are shown in Appendix A.

Next, beta-carotene PEM capsules were transfected in the D407 cells and a representative Raman map is depicted in Figure 6.

In order to avoid beta-carotene degradation upon exposure to laser light, very low laser power (mW) was used when imaging cells exposed to this type of capsule.

As shown in Figure 6, PC1 comprises the highest variability of the spectra, the loading plot showing a typical fluorescence background emitted by beta-carotene when excited with 532 nm laser. The Raman bands assigned to C-C (1157 cm^−1^) and C=C (1517 cm^−1^) stretching vibrations of beta-carotene backbone can be observed as well on the fluorescent background. Thus, the two structures observed in the PC1 score plot of Figure 6 are assigned to the beta-carotene loaded capsules.

The score plot in PC3 displays in blue color the cell shape, the beta-carotene capsules being observed in yellow-red colors. The negative bands in the loading plot, below 300 and at 1640 cm^−1^, are assigned to typical water vibrations. The red yellow-red color of the beta-carotene capsules is mainly due to the positive values in the spectral part higher than 1700 cm^−1^, due to the fluorescence pattern of beta-carotene.

The loading plot of PC4 shows clearly beta-carotene bands at 1153 and 1516 cm^−1^ in good concordance with the two beta-carotene capsules shown in the score plot of PC4. The PCA score and loading plots corresponding to the first ten PCs are shown in Appendix A.

So, the Raman imaging technique was able to discriminate the major cellular components, the PEMs, and the beta-carotene PEM capsules.

## 3. Materials and Methods

### 3.1. Materials and Reagents

The poly (allylamine hydrochloride), PAH, Mw 900,000 g/mol, and poly (sodium 4-styrene sulfonate), PSS, Mw 70,000 g/mol were purchased from Aldrich, dextran sulfate sodium salt MW 40,000 g/mol was purchased from AppliChem, CaCl_2_, and Na_2_CO_3_ from VWR, beta-carotene was supplied from Cayman Chemical Company (Ann Arbor, MI, USA), DMEM from Gibco.

### 3.2. Synthesis of Polyelectrolyte Multilayer Capsules

The PEM capsules were built around a CaCO_3_ core, following a previously reported protocol [25] (with some minor changes). The core serves as template only and, at the end, after all polymer layers were deposited, it was solubilized with EDTA. For the formation of the CaCO_3_ template core, equal amounts (6 mL) of 0.33 M CaCl_2_ and Na_2_CO_3_ were mixed under stirring at room temperature. Subsequently, 7.5 mL dextran sulfate 5 mg/mL was added (dextran is efficient in solubilizing the CaCO_3_ core at the final step). The polyelectrolyte layers were built around the CaCO_3_ cores using alternating layers of positively charged poly (allylamine hydrochloride), PAH, and negatively charged poly (4-styrene sulfonate), PSS, 2 g/L, 10 mL, prepared in 0.5 M NaCl. Three washing cycles with water (centrifugation, removing supernatant, addition of water) were carried out after the formation of each polymer layer. In total, six polymer layers were formed, followed by the dissolution of the CaCO_3_ core using 10 mL EDTA solution 0.2 M, pH 5.5. Eight washing steps in water were performed after core dissolution, by centrifugation at 120 g, 8 min. The final structure of the obtained PEM capsules is PSS/PAH/PSS/PAH/PSS/PAH. To obtain the beta-carotene loaded microcapsules, first a stock solution of beta-carotene (5 mg/mL in tetrahydrofuran) was prepared. Desired amounts of solubilized beta-carotene were mixed with the dextran solution and added after the core formation step.

### 3.3. Beta-Carotene Encapsulation Efficiency

The beta-carotene encapsulation efficiency was quantified using an Agilent HPLC instrument coupled with a DAD detector (Agilent Tehnologies, Santa Clara, CA, USA), using a reversed phase EC 250/4.6 Nucleodur 300-5 C-18 column (250 × 4.6 mm), 5 µm (Macherey-Nagel, Germany). The mobile phase consisted of mixtures of acetonitrile: Water (9:1, *v*/*v*) with 0.25% triethylamine (A) and ethyl acetate with 0.25% triethylamine (B). The gradient started with 90% A at 0 min to 50% A at 10 min. The percentage of A decreased from 50% at 10 min to 10% A at 20 min. The flow rate was 1 mL/min and the chromatogram was recorded at 450 nm [26]. The HPLC peaks were identified and quantified based on a calibration curve registered using freshly prepared beta-carotene standard solutions (Cayman Chemical, Ann Harbor, MI, USA). In short, sampling was carried out after each single step in the PEM microcapsules synthesis (after core formation, after adding each polymer, after every single washing step, after adding EDTA). This resulted in a total of 30 samples. From each of these aqueous samples, beta-carotene was extracted and quantified.

In short, 1 mL of ethyl acetate was used to extract beta-carotene from each aqueous sample. After extraction, the organic phase was collected, the solvent was evaporated, and the dried samples were kept at freezing until the next day, when the beta-carotene was again solubilized in ethyl acetate and 20 μL of the solution was injected in the HPLC column. The solvent evaporation was a necessary step since we have observed partial degradation of beta-carotene upon overnight storage if the solvent was present in the samples. Taking into account the beta-carotene instability in atmospheric conditions, the stock solution was also quantified by HPLC to have a correct estimation of the total amount that could be encapsulated. The nonencapsulated beta-carotene was quantified by summing up beta-carotene that resulted from each washing step. The difference between the initial beta-carotene amount and the amount lost during the synthesis represents the encapsulated beta-carotene. The nonencapsulated beta-carotene was calculated from the calibration curve with the equation y = 0.3961x + 6.6094, R^2^ = 0.9918. For the stock solution, the equation y = 1.6793x – 3280.7, R^2^ = 0.995 was used.

### 3.4. Cell Culture and Treatment

The human retinal epithelial D407 cell line was kindly donated by Prof. em. Dr.Dr.h.c. Horst A. Diehl, University of Bremen. The cells were cultured in Dulbecco’s Modified Eagle Medium supplemented with 10% fetal bovine serum, 1 mM sodium pyruvate, 100 U/mL penicillin, 100 mg/mL streptomycin, and 2.5 mg/mL amphotericin B. The cells were incubated at 37 °C, in 5% CO_2_ atmosphere, and 95% relative humidity. For the Raman mapping experiments the cells were grown on 35 mm sterile Petri dishes and seeded on MgF_2_ plates, as detailed in the section below.

### 3.5. Methods

#### 3.5.1. Raman Spectroscopy and Mapping

The Raman spectra were recorded with a Renishaw inVia Reflex Raman Spectrometer (Renishaw, New Mills, UK), equipped with an upright Leica microscope, an 1800 lines/mm grating, a CCD detector, and an automated step motorize stage to raster scan the sample area during the mapping experiments. The spectra were recorded using the 532 nm excitation laser line. The spectral resolution was 4 cm^−1^. Prior to spectral acquisitions, the wavelength was calibrated using an internal Si standard. The laser power during measurement was adjusted to either 120, 6, or 1 mW. Laser power of 120 mW was used for spectral acquisitions on samples containing empty PEM capsules, while for beta-carotene loaded capsules, adjustment of laser power to 6 or 1 mW was needed in order to avoid degradation of the beta-carotene molecule, which is heat and light sensitive. Spectra preprocessing consisted of cosmic ray removal.

MgF_2_ plates were used as solid support for the samples. MgF_2_ is a suitable support for Raman analysis, due to its extreme low fluorescent background. To record the Raman spectra on the PEM microcapsules, a drop of aqueous dispersion containing the capsules was deposited onto MgF_2_ plates and allowed to dry at room temperature. The samples were further analyzed under the Olympus 100X objective. The Raman maps were recorded by raster scanning the desired sample areas with 1 acquisition and 2 s exposure in each point. For imaging the nonloaded capsules, a 0.5 μm step size and 60 mW laser power were used, and for the beta-carotene loaded capsules 1 μm step size and 6 mW laser power were chosen. All data processing was carried out using custom-built MATLAB R2019b (The MathWorks Inc., Natick, MA, USA) functions. Cosmic ray spikes were removed from each spectrum before processing. For the identification of various cell components, PCA was performed.

To record Raman spectra on D407 cells upon exposure to different formulations of PEM capsules, the cells were grown for 24 h on sterile Petri dishes of 35 mm diameter (1.2 × 10^5^ cells seeded). The day after, the growth medium was replaced with PEM capsules diluted in the cell culture medium at a concentration of 10 capsules/cell, for another 24 h. Further, D407 cells were washed three times with phosphate buffer saline (PBS), detached from the Petri dishes using trypsin, centrifuged, and reseeded in Petri dishes with MgF_2_ plates inside. The PEM capsules treated cells were allowed to attach to the MgF_2_ plates for 24 h, followed by three times washing with PBS and cell fixation onto the plates. Cell fixation was carried out with 4% paraformaldehyde, at room temperature, for 20 min. After immobilization, the samples were again washed and stored in PBS buffer. The D407 cells exposed to PEM microcapsules were analyzed under the Olympus 60x water immersion objective. The maps were recorded with 2 μm step size, 1 s exposure time and 1 acquisition at each measuring point, and a laser power of 120 mW (samples exposed to nonloaded PEM capsules) or 6 mW (samples exposed to beta-carotene loaded PEM capsules).

#### 3.5.2. Multivariate Data Analysis

Data processing, including PCA and cosmic ray removal was carried out using custom-built MATLAB functions (MathWorks, Natick, MA, USA). PCA is a statistical method which reduces the dimensionality of the data by transforming the coordinate system of the data (in this case all Raman spectra from each pixel) into a coordinate system with less dimensions, while keeping as much of the variance of the data set. Therefore, the output of PCA is a set of spectra resembling plots, termed loading vectors, which correspond to the new coordinate system of the data set, and corresponding point-by-point mappings, termed score plots. Physically, the loading vectors represent the Raman bands that explain best the variance in the data set, i.e., how the Raman spectra of the sample in each pixel changes. Since loading vectors do not represent physically obtainable spectra but must be regarded as the coordinate system of the data, the interpretation of loading vectors must consider both positive and negative peaks. The score plot depicts visually the spatial correlation of the loading vector with the mapped sample. The biochemical structures displayed in the score plots can be assigned by identifying the biochemical information provided by the loading vector in the corresponding pixels [27].

Prior to PCA, cosmic ray removal was performed, which eliminates any artifact bands from the Raman spectrum. Since the PCA separates any redundant information from the Raman spectra (analogous to deconvolution), baseline removal was not necessary, since it appears only in one loading vector (due to the loading vectors being orthogonal with each other).

#### 3.5.3. Physical Characterization of PEM Capsules

The hydrodynamic diameter and surface charge of the beta-carotene loaded capsules were assessed using a Malvern Zetasizer Nano ZS equipped with a 4 mW He-Ne laser operating at 633 nm and an avalanche photodiode detector (Malvern Instruments, Worcestershire, UK). Zeta potential was measured by laser Doppler electrophoresis and the hydrodynamic diameter of the particles was measured by DLS. The measurements were carried out at 25 °C. Scanning electron microscopy (SEM) analysis of capsules were done on Hitachi SU8230 microscope (Hitachi, Krefeld, Germany) at 200 kV and 8 mm working distance.

The TEM experiments were carried out using a JEOL model JEM1010 (Jeol, Peabody, MA, USA) instrument operating at 80 kV accelerating voltage and equipped with a MegaViewIII CCD camera. For the TEM experiments, D407 cells were seeded for 24 h on Falcon cell culture inserts (0.4 μm) (Becton Dickinson Labware, Bedford, MA, USA), (8 × 10^3^ cells/cell). The cells were afterwards incubated for another 24 h with capsules (10 capsules/well). The culture medium was then removed, and the cells were washed three times with PBS and prefixed for 1 h with glutaraldehyde (2.5% in PBS). Next, three steps of rinsing with PBS and cells were post-fixed for 1 h in osmium tetroxide (2% in PBS). Dehydration was carried out in HPLC grade acetone in distilled water dilutions, followed by embedding with Epon resin. The resin was polymerized at 60 °C for 48 h, then cooled for 12 h. Ultrathin sections of about 70 nm, obtained on a diamond knife (Diatome) with Leica UC6 ultramicrotome (Leica Microsystems, Wetzlar, Germany) were post-stained with lead citrate and uranyl acetate. Sections collected on 200 mesh Cu grids were further examined.

## 4. Conclusions

In this study, we have obtained two formulations of polyelectrolyte multilayer capsules: Nonloaded formulations and microcapsules loaded with beta-carotene. Raman mapping was successfully used as a tool to investigate the polyelectrolyte multilayer capsules alone and in vitro, after uptake in D407 cells. Regarding the as-obtained microcapsule formulations, it was possible to distinguish between beta-carotene loaded vs. nonloaded capsules and to effectively map the beta-carotene distribution inside the microcapsules core. Moreover, after exposing the D407 cells to microcapsules, by analyzing Raman specific fingerprints of the cells and capsules, it was possible to localize the capsules intracellular information also confirmed by TEM.

## Figures and Tables

**Figure 1 molecules-25-01477-f001:**
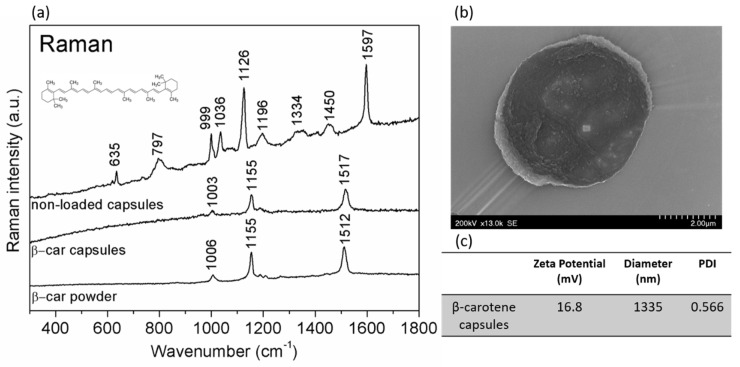
(**a**) From top to bottom, Raman spectra of empty polyelectrolyte multilayer (PEM) capsules, beta-carotene PEM capsules, and beta-carotene powder; (**b**) SEM image of one PEM capsule; (**c**) zeta potential values and hydrodynamic diameter of the beta-carotene PEM capsules.

**Figure 2 molecules-25-01477-f002:**
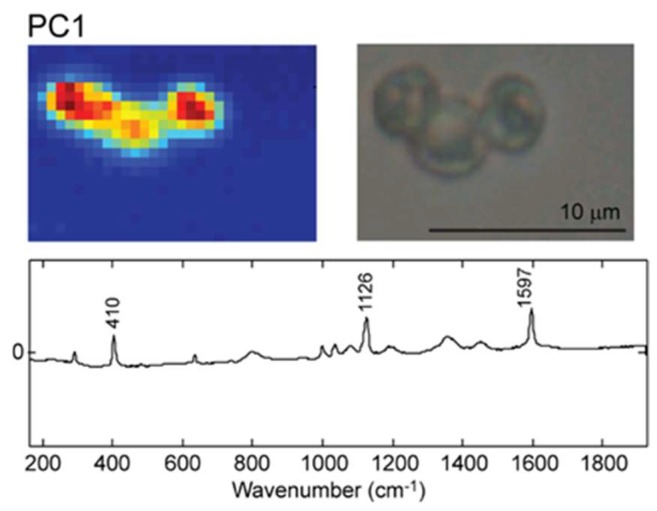
Top left: The false color Raman map of nonloaded microcapsules. Top Right: Optical microscopy image of nonloaded microcapsules. Bottom: The corresponding loading plot of the relevant principal component principal component 1 (PC1).

**Figure 3 molecules-25-01477-f003:**
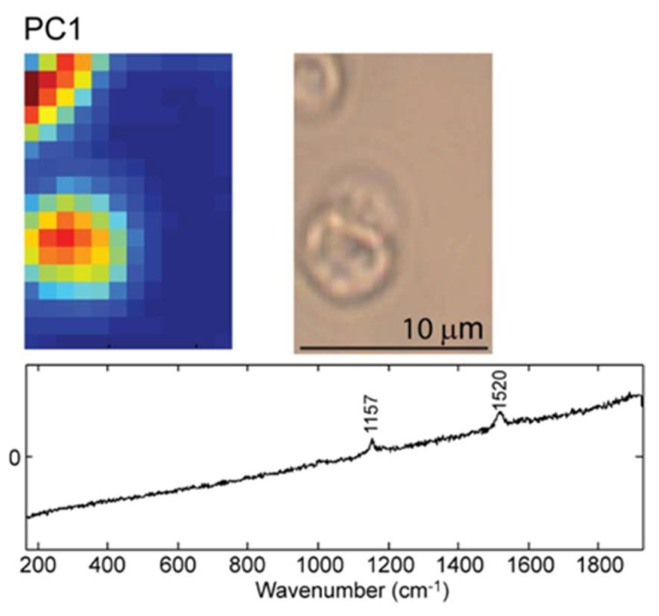
Top left: Raman map of beta-carotene loaded microcapsules. Top right: Microscopy image of beta-carotene microcapsules. Bottom: The corresponding loading plot of the relevant principal component PC1.

**Figure 4 molecules-25-01477-f004:**
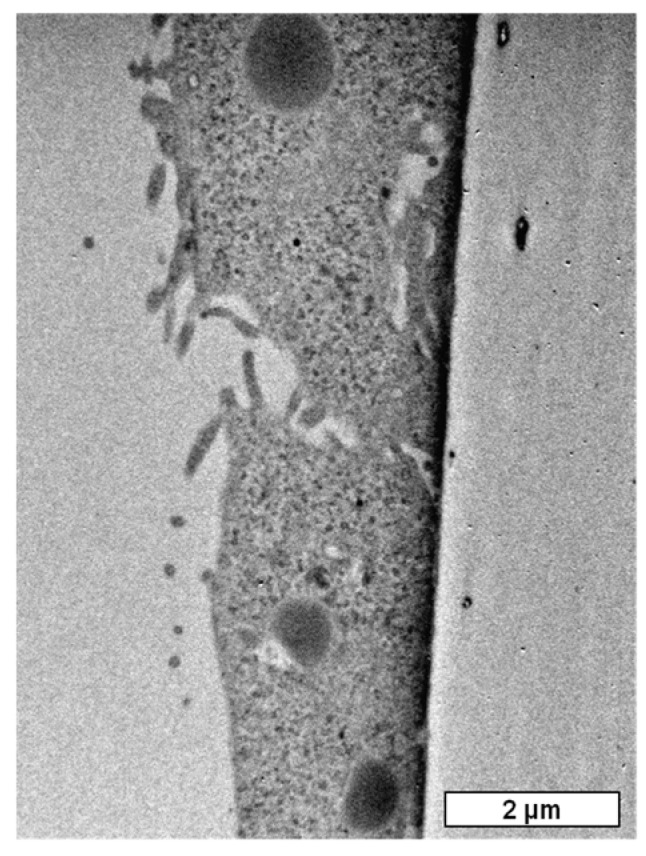
TEM micrograph of a D407 cell, after exposure to PEM microcapsules, showing three internalized microcapsules.

**Figure 5 molecules-25-01477-f005:**
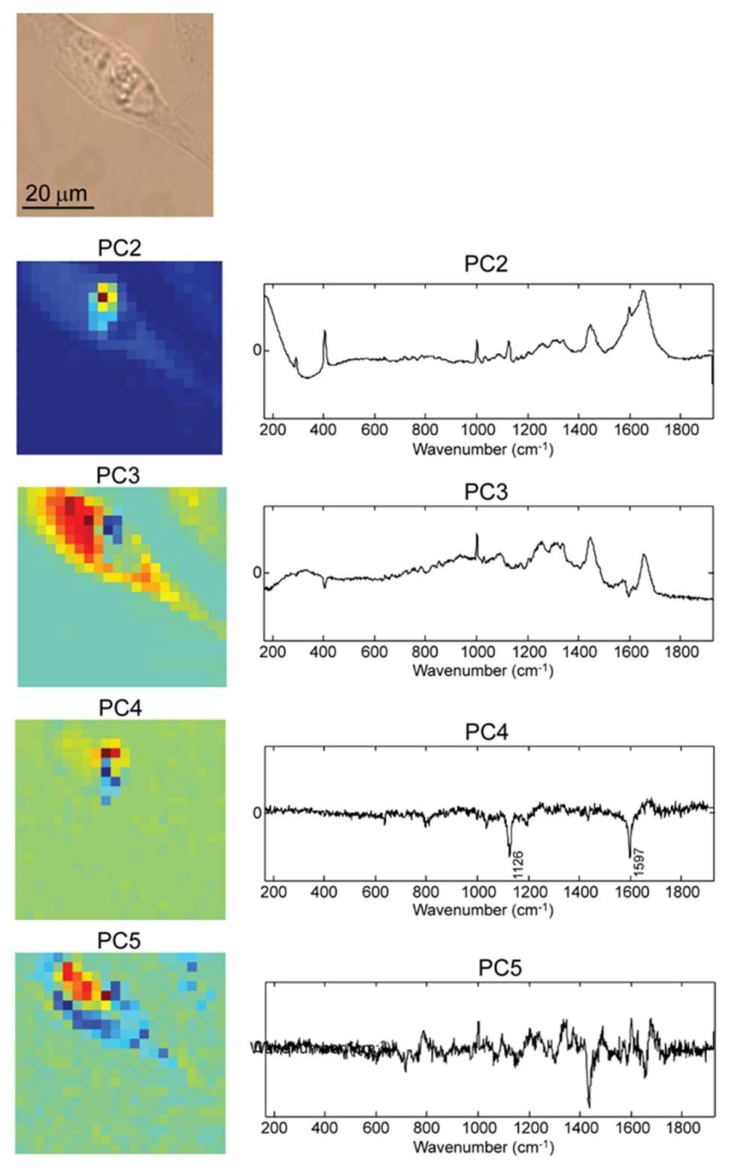
From top to bottom: Optical microscopy image of a D407 cell transfected with empty PEM capsules; followed by the Raman score plot (map) and the corresponding loading plot of relevant principal component PC2, PC3, PC4, and PC5.

**Figure 6 molecules-25-01477-f006:**
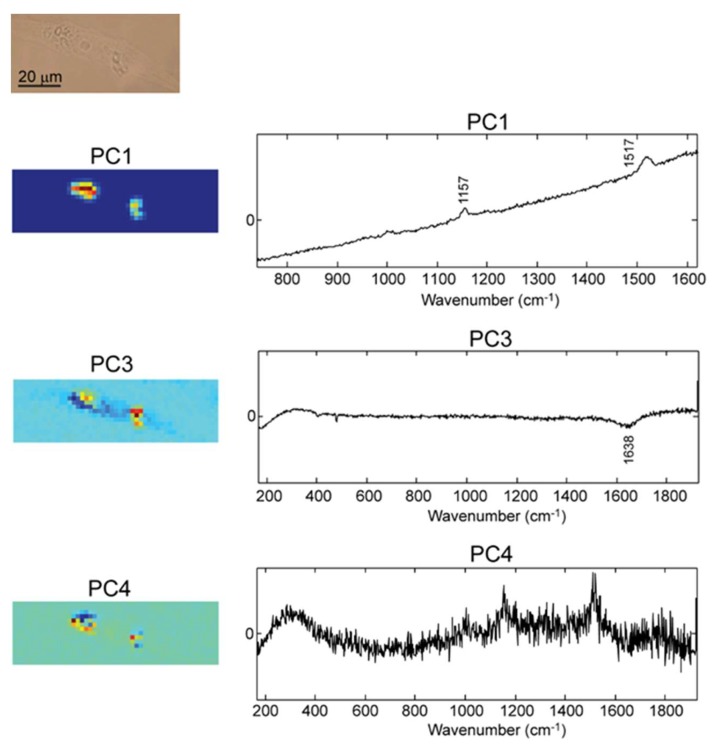
From top to bottom: Optical microscopy image of a D407 cell transfected with beta-carotene loaded PEM capsules followed by the Raman score plots (maps) and the corresponding loading plots of relevant principal components PC1, PC3, and PC4.

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
