# Peer review of "Cellular Internalization of Beta-Carotene Loaded Polyelectrolyte Multilayer Capsules by Raman Mapping"

_molecules, 2020, doi:10.3390/molecules25071477_

Round 1
Reviewer 1 Report
This article describes the preparation and characterization of beta-carotene containing polyelectrolyte multilayer microcapsules, and their uptake by human retinal epithelial cells, examined by Raman spectroscopy. The microcapsules were prepared by layer-by-layer adsorption method on calcium-carbonate cores, which were dissolved with a slightly acidic EDTA solution. The size, distribution, and zeta-potential of the microcapsules were characterized by scanning electron microscopy, dynamic light scattering, and laser Doppler electrophoresis, respectively. The beta-carotene content of the microcapsules was quantified by HPLC. The carotene-loaded microcapsules were successfully internalized into cells. The authors proved that Raman mapping is suitable to detect the presence of both empty and carotene-loaded capsules even in cells.
The manuscript is basically well presented, and the work is of high quality. Nevertheless, some minor modifications are recommended:
The chemical structure of beta-carotene, shown in the abstract, is not correct (it is actually an alpha-carotene). I would suggest giving the correct formula in the main text, as well, for example together with its Raman-spectrum.
In the introduction, the authors state: “However, the main natural sources of beta-carotene (yellow, orange and green leafy vegetables and fruits) are not an efficient way to provide the optimal daily dosage of carotenoids for a healthy diet [7]”. The cited reference No. 7. does not contain any indication supporting the above statement. Moreover, many publications describe the unambiguous correlation between high plasma levels of carotenoids and the consumption of carotenoid-rich fruits and vegetables. On the other hand, beta-carotene is one of the most stable and relatively cheap carotenoid, that is why it is widely used for coloration in the cosmetic and food industry. Its low water-solubility requires efficient solubilization methods, such as micro/nanocapsulation, for the above-mentioned applications. A novelty of the present research could be the development of a carrier for beta-carotene, which may be envisaged even for industrial applications.
In my opinion, the Materials and Methods section should precede the Results. It would be especially useful since the explanation of denotations (such as PSS, PAH, PEM) appears only in the M&M section, and the results would be much easier to understand.
The synthesis of the microcapsules is a modification of a method previously described, so a reference should be added.
It was not clear, if the physical parameters (size distribution, shape, zeta-potential) of the microcapsules were determined both for the empty and carotene-loaded capsules or only for the empty ones.
The beta-carotene content of the microcapsules was determined by HPLC method. I suggest describing the quantification method briefly, e.g. peak areas were compared with a calibration curve or external standard...
The release of beta-carotene from the microcapsules inside the cell was not describe. My question is whether this phenomenon was not observed or was not investigated? Anyway, it would be an interesting topic for future research. The fate of PSS/PAH microcapsule in cells, however, should be mentioned e.g. in the Introduction, if literature data about it is available.
Author Response
Please, see the attachment

Reviewer 2 Report
Leopold et al describe multilayered microcapsules loaded with beta-carotene. They supply these microcapsules to retinal epithelial cells and use Raman mapping to localize them. The authors create Raman images by dimensional reduction via PCA.
The paper is for the most part well written and easy to follow.
My main criticism is with how PCA is used:
- It appears that the authors developed a distinct PCA model for each image in their dataset. PCA captures the characteristic trends in each dataset, so when it is applied to different Raman images, different models will emerge, each only applicable to that specific image, making it impossible to compare different images.
- Additionally, how the PCA data is reported is not consistent, and is confusing. For example, the variance captured (%) by each component and residuals are never reported. Some components are completely ignored for some images (PC1 in Fig 5, PC2 in Fig 6). The authors need to provide all the data if the model is to be meaningful.
- Spectral preprocessing is needed if PCA is to be used. The authors assert on line 314 that no baseline removal is necessary. This assertion cannot be generalized, as a high non-specific background will artificially inflate the PC score and give false positive signals.
- Colorbars need to be added to the images. This is important to show which colors are positive numbers and which negative, which is meaningful for interpreting correlations with each PC.
I recommend the following:
Classical Least Square regression is a much more appropriate analysis method, where the spectra of loaded and empty capsules can be used as reference to provide images that are comparable between different scans and much easier to interpret.
Overall, I think the data and the description are interesting, and make a nice story. Once the authors present the data in a more coherent way (using PCA or a different technique) then the manuscript will be suitable for publication.
Minor points:
Line 82: What is the size-variation of the microcapsules? I see the polydispersity index is high (0.566). It would be useful to can provide the size distribution of the particles. In most pictures the microcapsules look larger than 2 µm in diameter.
Please define acronyms upon first use only: PSS, PAH, PEM, PCA
Author Response
Please, see the attachment

Reviewer 3 Report
The manuscript by L.F. Leopold et. al. titled “Cellular internalization of beta-carotene loaded polyelectrolyte multilayer capsules by Raman mapping” describes Raman mapping study of layer-by-layer encapsulated beta-carotene microcapsules and the possibility of their internalization into cells. It is a nicely done work, however, I have a couple of concerns listed below. My major concern is the conclusions.
Following are specific comments I recommend to address:
- Optical images of cells need to be matched width-wise with Raman maps so that correlation of positions could be made – Figures 5 and 6.
- Conclusions seem to be composed rather hastily. For example:
- “Moreover in vitro tests on D407 cells were able to localize the capsules in the cellular cytoplasm and to discriminate them with respect to and other cellular components.”
It is hard to understand with respect to what?
- “By using the technique of Raman mapping it was possible to distinguish between beta-carotene loaded vs non-loaded capsules”.
This has not been demonstrated – there was no experiment done with co-transfecting cells with both types of capsules and no differentiation was made.
Author Response
Please, see the attachment

Reviewer 4 Report
This paper presents a study reporting the production of polymeric capsules loaded with beta-carotene and their internalization into cells. Spatial resolved Raman spectroscopy has been performed on both microcapsules and cells, allowing to identify the signature of beta-carotene. The work presents an interesting concept, however, I believe that their data are not conclusive mainly because they show only a very limited number of cells and capsules and their corresponding Raman signatures. In this case, unfortunately, it is not possible to know if their observations are a real effect efficiently occurring in the cells investigated in their study or if it is only observed for a few cells and, thus, represent the exception and not the rule. The authors are advised to provide more data, with statistical significance, to proof that their observations and claims are effectively occurring in their cultured cells. In addition, some improments in the presentation of the articles are recommended:
Introduction, pg.2, lines 51-52. Authors state that natural sources of beta-carotene (essentially vegetables largely used in nutrition) are not an efficient way to provide optimal daily dosage of carotenoids. This should be better explained to let it clear to the reader why a healthy diet based on the mentioned vegetables and fruits is not sufficient to achieve the proper daily dosage of carotenoids.
SEM image corresponds to only one particle, which is found to be much higher than the average sizes revealed by DLS (including the PDI). How representative are this image from their sample? Authors are advised to provide more SEM images, if possible, showing different magnifications to support their claims on the state of their surfaces and sizes. To simply provide a direct space visualization of their particles, maybe optical microscopy could be sufficient just to show their particles. The valuable information from SEM should be finer details at the surface of the particles.
Raman spectra in Fig. 1a should be provided with the corresponding Y-scales. The authors state that vibrations from non-loaded capasule do not contribute for the final spectra of beta-car containing particles; however, it is not clear to the reviewer if the absence of peaks from capsules arises from the fact the signal that their signal is to weak compared to the loaded particles or if some chemical modification occurred. Considering the level of noise in beta-car capsule spectrum, I guess that signal from this formulation is much weaker than the signal from non-loaded capsules. In this case, are the authors sure that capsules are present in their samples? Can phase separation issues affect your measurements?
Figures 2 and 3. The wavenumbers corresponding to the maps should be clarified in the images. For instance, do the map shown in Fig. 2 correspond to absorbance at 1126 cm-1? At 1597 cm-1? To the integrated Raman signal? The same question applies to Fig. 3: it is not clear to this reviewer the wavenumber (s) to which the map corresponds to.
Round 2
Reviewer 2 Report
Leopold et al have made some improvements to their work.
I concede that PCA can be used to analyze each image/dataset independently.
However, I maintain that presenting a PCA model partially is really not useful for understanding the study. The authors should include the loadings for every model, with variance captured. This of course can be included in the SI (is there an SI file? I didn’t see one).
On the point of spectral preprocessing: PCA requires mean-centered data, in order to extract meaningful covariance between the variables. By including the fluorescence peak (without baseline removal) the dataset is biased heavily. This is apparent in the plots provided in the response to my previous critique, showing the variance captured, where the first PC expresses almost all of the dataset’s variance.
If this (using the fluorescent background) is crucial to the analysis and not an artifact it should be discussed in the text. This comes back to not showing all the PCs. I cannot assess whether the fluorescence is helping or hurting the analysis.
On the matter of the colorbars. I understand that quantification is not the purpose of this study, and without units on the loadings it makes no sense to use units on the maps. However, since PCA is used, it is important to know where in the image we have correlations with the PC and where counter-correlations. I.e. what color is zero?
Once these points are addressed in the manuscript (and SI) the manuscript will be suitable for publication.
Author Response
Leopold et al have made some improvements to their work.
I concede that PCA can be used to analyze each image/dataset independently.
However, I maintain that presenting a PCA model partially is really not useful for understanding the study. The authors should include the loadings for every model, with variance captured. This of course can be included in the SI (is there an SI file? I didn’t see one).
According to the Reviewer’s suggestion, we prepared a SI file where we included the first ten score and loading plots for all Raman mapping figures shown in the manuscript (Figure 2, 3, 5 and 6). Each figure contains also a graphical representation of the explained variance of the first ten PCs. As can be observed from the graphical representations, in each case the cumulative explained variance was over 90%.
On the point of spectral preprocessing: PCA requires mean-centered data, in order to extract meaningful covariance between the variables. By including the fluorescence peak (without baseline removal) the dataset is biased heavily. This is apparent in the plots provided in the response to my previous critique, showing the variance captured, where the first PC expresses almost all of the dataset’s variance.
If this (using the fluorescent background) is crucial to the analysis and not an artifact it should be discussed in the text. This comes back to not showing all the PCs. I cannot assess whether the fluorescence is helping or hurting the analysis.
According to the Reviewer’s suggestion, a discussion in the manuscript text regarding the fluorescent background of beta carotene was included. On page 5, after Figure 3, the following discussion was added:
The first PC of the Raman spectra PCA provides a Raman map with very good spatial resolution, enabling the identification of the beta-carotene loaded PEM capsules. The corresponding PC1 loading plot shows the two beta carotene marker bands at 1157 and 1520 cm-1, as well as the beta-carotene typical fluorescent background.
Also on page 7, after Figure 6, the following discussion, which mention also the fluorescent background of beta carotene as marker feature for its detection:
As shown in Figure 6, PC1 comprises the highest variability of the spectra, the loadings plot showing a typical fluorescence background emitted by beta-carotene when excited with 532 nm laser. The Raman bands assigned to C-C (1157 cm-1) and C=C (1517 cm-1) stretching vibrations of beta-carotene backbone can be observed as well on the fluorescent background.
On the matter of the colorbars. I understand that quantification is not the purpose of this study, and without units on the loadings it makes no sense to use units on the maps. However, since PCA is used, it is important to know where in the image we have correlations with the PC and where counter-correlations. I.e. what color is zero?
According to the Reviewer’s suggestion, all the score plots regarding the Raman mapping figures (Figure 2, 3, 5 and 6) now contain colorbars, which were automatically generated. The corresponding figures containing the colorbars were inserted in the SI file (Figures S1-S4). We would sincerely like to thank the reviewer for all the useful and constructive comments. Wereally hope that addressing these comments lead to meaningful improvements to our manuscript.
Once these points are addressed in the manuscript (and SI) the manuscript will be suitable for publication.
Reviewer 3 Report
The authors have not addressed my concerns and did not make any attempts to make corrections.
Author Response
The manuscript by L.F. Leopold et. al. titled “Cellular internalization of beta-carotene loaded polyelectrolyte multilayer capsules by Raman mapping” describes Raman mapping study of layer-by-layer encapsulated beta-carotene microcapsules and the possibility of their internalization into cells. It is a nicely done work, however, I have a couple of concerns listed below. My major concern is the conclusions.
Following are specific comments I recommend to address:
- Optical images of cells need to be matched width-wise with Raman maps so that correlation of positions could be made – Figures 5 and 6.
The optical images were provided by us in order to make it possible to easily observe the correspondence between the Raman maps and optical microscopy images. They were added as additional proof, but it was not our purpose to obtained width-wise correlation of the positions, the resolution of the Raman mappings being in anyway different, in the order of 1 µm.
- Conclusions seem to be composed rather hastily. For example:
- “Moreover in vitro tests on D407 cells were able to localize the capsules in the cellular cytoplasm and to discriminate them with respect to and other cellular components.”
It is hard to understand with respect to what?
This has not been demonstrated – there was no experiment done with co-transfecting cells with both types of capsules and no differentiation was made.
According to reviewer 3 suggestions, from which the conclusions were considered the major drawback of our manuscript, we tried to reformulate the Conclusions paragraph, in our attempt to render this it more clear. We hope the new version suits better. The new version of the manuscript now contains the updated Conclusion section, which is also provided below:
In this study, we have obtained two formulations of polyelectrolyte multilayer capsules: non-loaded formulations, and microcapsules loaded with beta-carotene. Raman mapping was successfully used as a tool to investigate polyelectrolyte multilayer capsules alone and in vitro, after uptake in D407 cells. Regarding the as-obtained microcapsule formulations, it was possible to distinguish between beta-carotene loaded vs non-loaded capsules and to effectively map the beta-carotene distribution inside the microcapsules core. Also, after exposing the D407 cells to microcapsules, by analyzing Raman specific fingerprints of the cells and capsules, it was possible to localize the capsules intracellular, information also confirmed by TEM.
Reviewer 4 Report
The authors responded most of my comments appropriately. Only one more point remains to be improved:
« We did not consider of importance to show all the maps that we have, since the results are consistent from one map to another for the same type of samples. »
This reviewer believes that it is important to present the data to proof that their findings are consistent. Since the authors already have maps from independent assays showing consistency between samples, why not to show them (in a SI file, for instance)? In this case, they are advised to include a few more maps to let it undoubably clear to the reader about robustness of their findings.
Author Response
The authors responded most of my comments appropriately. Only one more point remains to be improved:
« We did not consider of importance to show all the maps that we have, since the results are consistent from one map to another for the same type of samples. »
This reviewer believes that it is important to present the data to proof that their findings are consistent. Since the authors already have maps from independent assays showing consistency between samples, why not to show them (in a SI file, for instance)? In this case, they are advised to include a few more maps to let it undoubably clear to the reader about robustness of their findings.
The SI info file now contains additional mappings for the analyzed samples (Figures S5-S10). One additional mapping for both non-loaded capsules (Fig S5) and beta carotene loaded capsules (Fig S6) and two more maps for both D407 cells exposed to non-loaded capsules (Figs. S7 and S8) and D407 cells exposed to beta-carotene loaded capsules (Figs. S9 and S10).